# Novel Fluticasone Propionate and Salmeterol Fixed-Dose Combination Nano-Encapsulated Particles Using Polyamide Based on L-Lysine

**DOI:** 10.3390/ph15030321

**Published:** 2022-03-08

**Authors:** Mohammad H. Alyami, Eman Zmaily Dahmash, Dalia Khalil Ali, Hamad S. Alyami, Hussien AbdulKarim, Samar A. Alsudir

**Affiliations:** 1Department of Pharmaceutics, College of Pharmacy, Najran University, Najran 55461, Saudi Arabia; mhalmansour@nu.edu.sa; 2Department of Applied Pharmaceutical Sciences and Clinical Pharmacy, Faculty of Pharmacy, Isra University, Amman 11622, Jordan; eman.zmaily@iu.edu.jo (E.Z.D.); hussein.abdalkarem@yahoo.com (H.A.); 3Department of Physiotherapy, Faculty of Allied Medical Sciences, Isra University, Amman 11622, Jordan; dalia.ali@iu.edu.jo; 4National Center of Biotechnology, Life Science and Environmental Research Institute, King Abdulaziz City for Science and Technology, Riyadh 11442, Saudi Arabia; salsadeer@kacst.edu.sa

**Keywords:** fluticasone propionate, salmeterol xinafoate, polyamide, respiratory drug delivery, L-lysine, nanocapsules

## Abstract

One of the key challenges in developing a dry powder inhaler (DPI) of an inhalable potent fixed-dose combination (FDC) is the ability of the formulation to generate an effective and reproducible aerosol able to reach the lower parts of the lungs. Herein, a one-step approach is presented to expedite the synthesis of nanoaggregates made from a biocompatible and biodegradable polyamide based on L-lysine amino acid employing market-leading active pharmaceutical ingredients (fluticasone propionate (FP) and salmeterol xinafoate (SAL)) for the management of asthma. The nanoaggregates were synthesized using interfacial polycondensation that produced nanocapsules with an average particle size of 226.7 ± 35.3 nm and zeta potential of −30.6 ± 4.2 mV. Differential scanning calorimetric analysis and x-ray diffraction, as well as scanning electron microscopy of the produced FDC, revealed the ability of the produced nanocapsules to encapsulate the two actives and display the best aerodynamic performance. The FDC nanocapsules displayed 88.5% and 98.5% of the emitted dose for FP and SAL, respectively. The fine particle fraction of the nominated dose was superior to the marketed product (Seretide Diskus^®^, Brentford, United Kingdom). The in-vitro release study showed an extended drug release profile. Our findings suggest that nanoaggregates using polyamides based on L-lysine and interfacial polycondensation can serve as a good platform for pulmonary drug delivery of FDC systems.

## 1. Introduction

The use of a fixed-dose combination (FDC) containing long-acting β2 agonists (LABAs), such as salmeterol xinafoate (SAL), and inhaled corticosteroids (ICSs), such as fluticasone propionate (FP), for the management of asthma and other respiratory conditions is recommended [1]. FDCs simplify the treatment regimen and therefore enhance patients’ adherence. Further, FDCs have the potential to improve and prolong the effect of monotherapy. In particular, the FDC of long-acting β2 agonist (LABA) and inhaled corticosteroid (ICS) is considered as a key approach for the management of asthma and patients with severe chronic obstructive pulmonary disease (COPD), particularly those experiencing exacerbations [2,3,4,5]. The National Asthma Education and Prevention Program Coordinating Committee Expert Panel Working Group updated the Asthma Management Guidelines in 2020. The group recommendations were focused on using combined medium to high dose ICS-LABA therapy for the management of persistent asthma in children ages 0–4 years, high dose ICS-LABA for children ages 5–11 years, medium to high dose of ICS-LABA for individuals ages 12+ years [5]. The Food and Drug Administration (FDA) produced a safety report about the findings of four large clinical trials to assess the efficacy and safety of the combined use of ICS with LABA. The results showed that the ICS/LABA combination reduced asthma exacerbations compared to ICS alone [6]. Furthermore, asthma is one of the main non-communicable diseases, with a global prevalence exceeding 339 million people and is most commonly prevalent among children [7,8,9]. According to the WHO, annual asthma-associated worldwide deaths exceeded 400,000 [8,9].

The development of dry powder inhaler (DPI) formulations is based on two approaches. The first is based on carrier-drug formulation using mainly inhalation grade lactose, which is the only Food and Drug Administration (FDA)-approved carrier for dry powder inhalation. However, lactose carrier-based DPI formulations are generally not recommended for patients with a known lactose allergy. Furthermore, the development of carrier-based formulation is challenging in terms of the mixing process and lactose characteristics. Furthermore, due to the low dose of ICS and LABA, content uniformity is a key challenge in this approach [10,11,12,13,14]. Therefore, DPI formulations using alternative novel approaches are deemed necessary.

The second approach is based on preparing nano or micro aggregates as carriers to deliver pulmonary drugs to the lung [15]. Many carriers are used, such as lipids and polymers. Most polymers used for pulmonary drug delivery are poly(DL-lactide-co-glycolide acid) (PLGA) [16,17,18], poly(lactic acid) (PLA) [19,20], and chitosan [21,22]. Among the different classes of synthetic biodegradable and biocompatible polymers, α-amino acids-based polyamides gained particular importance in the biomedical field due to the lower toxicity of their degraded products and excellent mechanical and thermal properties property [23,24]. Lysine amino acid is a basic, charged (at physiological pH), and aliphatic amino acid. It can be used as a diamine to react with suitable monomers to prepare a nonpeptidic polyamide. This polyamide is free from some of the disadvantages, for example, decomposition in the molten state and swelling in moisture environments of conventional poly(amino acid)s resulting from the repeating peptidic amide bonds [25].

Various approaches were employed in the development of nanoaggregates for DPI. Each offers advantages and limitations. For example, spray drying, and supercritical fluid technology, produce engineered particles with the desired particle size range. Nevertheless, the process requires a lot of optimization and often results in low yield [26,27,28]. Further, the two approaches are increasingly in use in the development of DPI formulation of higher doses but possess limited operating windows and cannot be employed for active pharmaceutical ingredients (APIs) that are heat sensitive [29].

The success of DPI use depends on the ability of the formulation to deliver the required dose of the API to the targeted sites in the respiratory system with few side effects [30]. The carrier-based formulation requires the API particles to detach from the surface of the carrier to reach the lower parts of the lungs. Nevertheless, nano or micro aggregates are made of homogenous dispersions of the API with or without polymers or other materials. Upon inhalation, the aggregates become fragmented into an aerolizable size (1–5 µm) that enables their deposition in the lower parts of the respiratory system. Then, the aggregates will disintegrate into their nano or submicron particles [11,12,31].

This project aims to develop DPI to deliver FP and SAL nanoparticle agglomerates as a single FDC. FP and SAL are the most used ICS and LABA, respectively [5,6]. The study illustrates the formulation of a polyamide based on L-lysine encapsulating both FP and salmeterol nanocapsules using a cost-effective method, interfacial polymerization, and compares the results to the commercially available FDCs of the two actives.

## 2. Results and Discussion

The development of a DPI using nanoaggregates required the production of aggregates that have sufficient aerodynamic performance, yet upon departure from the inhaler, disintegrate into smaller particles, preferably within 1–5 µm to enable deposition into the lower part of the respiratory system [12]. Owing to the small dose of the two actives used (250 µg for FP and 50 µg for SAL), there is a need to prepare the aggregates with the aid of biocompatible, biodegradable material such as polyamides. The synthesis of the FP and SAL loaded nanocapsules was carried out in a stepwise manner. After the completion of the polymer synthesis and characterization, individual API was introduced in the polymerization process. Then quantification of encapsulation efficiency for each API was commenced, which was followed by synthesis of the (FP-SAL/Lys-PA) NCs in a one-step process.

### 2.1. Synthesis of Polyamide from Lysine and 2,5-Furyldichloride (Lys-PA)

A polyamide based on lysine was prepared by interfacial polycondensation from the reaction of 2,5-furyl dichloride with lysine α-amino acid. In this technique, the polymerization process happens at the interface between the aqueous layer containing the lysine and the organic layer containing the 2,5-furyl dichloride. The vigorous stirring guarantees the two layers of dispersion. The interface refreshment increases the surface available for the reaction; moreover, the vigorous stirring facilitates the formation of fine particles of the polymer with reduced reactants and solvents encapsulation inside the polymer chain (Figure 1 illustrates the process).

### 2.2. Fluticasone Propionate and Salmeterol Xinafoate Polyamide Formula Synthesis ((FP-SAL/Lys-PA) NCs)

The interfacial polycondensation procedure was also employed for the synthesis of fluticasone propionate and salmeterol xinafoate polyamide nanocapsules based on L-lysine. The formed polyamide formula was a light-flowing powder containing FP and SAL for the delivery as a DPI by inhalation. This direct method of encapsulation of FP and SAL resulted in nanocapsules with an average diameter of around 199.3 nm and entrapment as well as encapsulation efficiency, as reported in Section 2.9.

### 2.3. Fourier Transform Infrared (FTIR) Spectroscopy Analysis

The FTIR spectra exhibit characteristic absorption bands for all organic functional groups of FP and SAL, as shown in Figure 2a,b. The FTIR spectrum of FP showed three characteristic sharp bands that represent (C=O) groups at 1744 cm^−1^ corresponding to the ester carbonyl group, 1701 cm^−1^ corresponding to the thioester carbonyl, and 1661 cm^−1^ to the ketone carbonyl, as shown in Figure 2a. The FTIR spectrum of SAL showed the characteristic sharp bands of C-O at 1579 cm^−1^, 3319 cm^−1^ band of O-H, and 3024 cm^−1^ band to N-H Figure 2b. The FTIR spectrum of Lys-PA revealed two carboxylic acid characteristic infrared stretching absorption bands. The C=O showed a stretching vibration at 1688 cm^−1^, and the O-H appeared as a broad band from 3325 to 2449 cm^−1^. On the other hand, in the amide group, a strong stretching vibration band for the formed amide group C=O was observed at around 1549 cm^−1^, and IR bands of the N-H bond appeared at 3149 cm^−1^ for the stretching at 1602 cm^−1^ for bending vibrations, as shown in Figure 2c. The comparison between (FP-SAL/Lys-PA) NCs with Lys-PA FTIR spectra shows the same characteristic bands. This may be due to the low loading of FP and SAL or overlapping between the polymer and drug bands, as shown in Figure 2d [23,24,32].

### 2.4. Nuclear Magnetic Resonance (NMR) Spectroscopy

#### 2.4.1. The ^1^H-NMR Spectrum of the Ly-PA

The CH proton of the lysine unit was observed at 4.38 ppm. The protons of four CH_2_ were observed at 3.05, 3.35, 1.87 and 1.23. The protons of NH were observed at 8.62 and 7.77 ppm. The OH proton of carboxylic acid was observed at 12.83. The proton peaks of the furan ring were observed in the range 7.28–7.08 ppm, as shown in Figure 3a.

#### 2.4.2. The ^13^C-NMR Spectrum of the Ly-PA

The formed carbonyl carbons of the amide bond appeared at 159.3 and 157.8 ppm, and the carbon of carboxylic acid appeared at 173.8 ppm. The CH of the furan ring was observed at 118.6 and 115.4 ppm, and the quaternary carbons at 149.0 and 147.5 ppm. The chemical shift of the carbon of the CH attached to the C=O of the carboxylic acid group in the Ly-PA was observed at 52.5 ppm. The four CH_2_ carbons were observed at 70.2, 38.7, 30.5 and 23.1 ppm, as shown in Figure 3b.

#### 2.4.3. 2D-NMR Spectra of the LY-PA

The DEPT 135-^1^H HMBC NMR spectrum demonstrates all correlations that exist in the CH_2_ and CH carbons and each of the peaks of the protons in Ly-PA, as shown in Figure 3c.

### 2.5. Particle Size, Zeta Potential and SEM Analysis

The average particle size, PDI and zeta potential of the APIs, the polymer and the API loaded polymer are summarized in Table 1 and Appendix A. It is noted that the actives (FP and SAL) as a powder demonstrated aggregated fine powder with a negative charge. The polymer demonstrated a particle size of around 199.3 nm and a negative charge that exceeded −30 mV. The high negative charge of the polymer is attributed to the presence of the carboxylic acid group, and the magnitude of the charge will support the de-aggregation of the particles. However, a slight but insignificant increase (*t* test, *p* > 0.274) of the particle size was observed upon the addition of FP and SAL, which attained a negative charge that will aid the de-agglomeration process.

The SEM images of the product, as depicted in Figure 4, showed aggregated particles for the FP-SAL NC. Successful respiratory DPI requires the presence of aggregates within a particle-size range of 1–5 µm, which enables efficient aerosolization and deposition of DPI formulation into the lower parts of the respiratory system. From Figure 4a, it can be noted that most particles are within this range (average aggregate size using ImageJ software was 3.52 ± 1.21 µm, *n* = 25 aggregates). Upon deposition, the aggregates will disintegrate into their primary nanocapsules. (FP-SAL/Lys-PA) NCs will settle at the lower part and provide a large surface area within the region for the release of both FP and SAL.

### 2.6. Differential Scanning Calorimetry (DSC) Analysis

FP and SAL had a melting point of 261–273 and 120–169 °C, respectively [33,34]. The DSC thermograms of the FP are shown in Figure 5a. From the graph, the melting point of FP occurs after the measurement window of the device (up to 250 °C) and hence was not identified. The thermal behavior of SAL (Figure 5b) demonstrated two endothermic peaks at 125.12 and 140.67 °C, which represent the melting points of the commonly reported two crystal forms of SAL (Form I and Form II) [22,35]. Further, the Lys-PA thermogram depicted in Figure 5d shows an endothermic band that started at 70.2 °C and peaked at 88.97 °C, which represents the glass transition of the polymer. The (FP-SAL/Lys-PA) NCs thermogram (Figure 5c) shows the broadened endothermic band at 92.75 °C. However, the broad band demonstrated an increase when compared with the Lys-PA and an increase in enthalpy, which could be attributed to the encapsulation of the FP and SAL in the polymer matrix [36]. Despite the low quantity of FP and SAL in the final formula, it is possible that FP and SAL transformed into amorphous material within the nanoparticles. Similar findings were reported by Amasya et al. [37].

### 2.7. X-ray Diffraction Analysis

XRD was performed to analyze the produced nanocapsules. The XRD patterns showed sharp diffraction peaks, suggesting that both FP and SAL were crystalline (Figure 6a,b) [38,39,40]. The pattern of the polymer showed some features of crystallinity as there were clear peaks at ~18°, 25°, 29° and 40°. The same peaks were evident in the pattern of the (FP-SAL/Lys-PA) NCs. The peaks at 18° and 25 overlap with FP and SAL peaks. Furthermore, due to the very low content of FP and SAL in the final NC, it is not possible to determine the change of crystallinity of the FP and SAL upon inclusion within the polymer.

### 2.8. HPLC Method for Quantification of Thymoquinone

Concurrent quantification of FP and SAL was validated according to the ICH guidelines for analytical method validation [41]. The retention time for SAL and FP was 5.20 ± 0.08 min and 8.34 ± 0.066 min, respectively, and the polymer components did not interfere with the APIs’ peaks. The method validation parameters are summarized in Table 2, indicating a valid and reproducible method.

### 2.9. Pulmonary Application of the Nanocapsules

The targeted FDC of FP and SAL was 250 and 50 µg per dose, respectively. However, to enable accurate dosing, the APIs were incorporated into the polymer with a targeted dose weight to range between 14 and 25 mg. Therefore, initial attempts were made to develop an individual API with the polymer, calculate its *EE*% and *ECE*%, then optimize the final formulation. Table 3 summarizes the mono and FDC formulas, their *EE* and *ECE*, as well as the recommended unit dose weight. Adjustments were made to the organic phase and aqueous phase volumes to ensure the APIs dissolved properly. F1 produced good powder, but the total recommended dose to supply the recommended dose was high (24.3 mg+ 7.9 mg = 32.2 mg); therefore, an attempt to increase the FP was made as in F2, which produced a recommended dose of 22.2 mg/dose. A one-step process was made in F3 by the reduction in SAL from 5 to 2 mg, and increasing the FP to 20 mg resulted in an acceptable weight of 15 mg. This weight in each dose can be easily weighed, and in terms of inhalation, it is within the acceptable range.

### 2.10. In Vitro Assessment of Aerodynamic Performance

The final formulation (F3) was further used for the assessment of the in vitro performance using the NGI. The results for the four inhalation parameters: (%ED), the FPF of the emitted dose (%FPF-ED), the RD) and the FPF from the nominated dose (FPF-ND) were assessed, as can be seen in Figure 7. The results revealed that the (FP-SAL/Lys-PA) NCs demonstrated an aerodynamic behavior that is superior to the marketed product (Seretide Diskus^®^) for FP.

Furthermore, the aerodynamic behavior of F3 and the marketed product for SAL is summarized in Figure 8. F3 showed superior results when it compared to the marketed product.

The comparison between the marketed product and the prepared formula revealed the ability of the nanoaggregates to effectively depart the aerolizer and travel through the respiratory system (ED) to an extent that is higher than the marketed product, which is based on carrier formulation.

The main parameter that is of high importance is the RD, which represents the cumulative amount of the API that will settle in the lower parts of the respiratory system and the FPF-ND, which represents the percentage of the given dose that reaches the lower part of the lungs. F3 demonstrated higher deposition into the lower parts of the lung.

The MMAD values for F3 and the marketed product for each active are presented in Table 4. From the table, it is noted that the MMAD values for the aggregated nanocapsules and the marketed product showed relatively similar results. Moreover, the GSD revealed that the two formulations produced particles with a similar spread of particle size (GSD).

### 2.11. Release Study

The release of both FP and SAL from the (FP-SAL/Lys-PA) NCs, as can be seen from Figure 9, demonstrated an initial burst effect of the two actives within the first two hours, which was higher for SAL; however, the release continued after that in a slow extended-release pattern. The reason for the burst effect is related to the quick dissolution of the FP or SAL that are attracted to the surface of the polymer. The extended release is attributed to the release of the drug particles from the nanocapsules. The polymer will undergo biodegradation thus enabling the drug to be released into the surrounding media.

The release profile was modeled using five mathematical models that can describe the kinetics of the release profile of FP and SAL. Appendix A as well as Appendix A show the linear regression lines of each mathematical model for FP and SAL, respectively. Furthermore, Table 5 summarizes the model constants and the coefficient of determination for each active. From the table, the release profile for both FP and SAL fits the Higuchi model, which represents the release of both FP and SAL by diffusion [42]. However, an understanding of the diffusion pattern of the two actives can be determined from the Korsmeyer–Peppas model, which showed the high correlation. The release pattern is governed by the value of “*n*”. For FP, the “*n*” value was 0.5541, which represents non-Fickian transport or anomalous transport. In this case, the mechanism of FP release is based on both diffusion of the drug as well as polymer swelling/relaxation and rearrangement of the polymer chains, resulting in the anomalous effects [43,44]. As for SAL, the release mechanism as per the “*n*” value (0.3176) is based on Fickian diffusion [44]. It is noted that the amount of SAL is low and hence it is expected to undergo diffusion. A similar release profile was reported in the literature for FP loaded in solid lipid nanoparticles [45].

## 3. Materials and Methods

### 3.1. Materials

FP, salmeterol xinafoate (SAL) and diethylene glycol were obtained from Sigma (Pool, UK). L-lysine and 2,5-furyl dichloride were purchased from Acros Organics (Geel, Belgium). Ethyl acetate, trifluoroacetic acid (TFA), dimethyl sulfoxide (DMSO), acetone, and acetonitrile were obtained from Alpha Chemika (Mumbai, India). Glycerin was obtained from Labchem (Zelienople, PA, USA). Additionally, potassium hydroxide (KOH) pellets were obtained from Riedel-de Haën (Seelze, Germany), while methyl alcohol, diethyl ether, and HPLC-grade water were purchased from Tedia High Purity Solvents (Fairfield, CT, USA).

### 3.2. Synthesis of Polyamide from Lysine and 2,5-Furyldichloride (Lys-PA)

A mixture of L-lysine (1.46 g, 10 mmol) and KOH (1.12 g, 20 mmol) was dissolved in distilled water (30 mL). Next, a 2,5-furyl dichloride solution (1.93 g, 10 mmol), dissolved in CHCl_3_ (10 mL), was added dropwise to the aqueous solution under vigorous stirring. The reaction was allowed to proceed under vigorous stirring for 30 min. The formed polymer was filtered off and washed several times with distilled water and diethyl ether. The last traces of solvent were removed by keeping the polyamide in a freeze dryer for six hours.

### 3.3. Fluticasone Propionate and Salmeterol Xinafoate Polyamide Formula Synthesis ((FP-SAL/Lys-PA) NCs)

A mixture of L-lysine (1.46 g, 10 mmol) and KOH (1.12 g, 20 mmol) was dissolved in distilled water (30 mL). Next, a 2,5-furyl dichloride solution (1.93 g, 10 mmol) was dissolved in CHCl_3_ (10 mL). FP (10 mg, 0.02 mmol) for F1, (20 mg, 0.04 mmol) for F2 and F3, and salmeterol xinafoate (SAL) (5 mg, 0.083 mmol) for F1 and F2, (2 mg, 0.0033 mmol) for F3 was added to the CHCl_3_ mixture using vigorous stirring. The CHCl_3_ solution was added dropwise to the aqueous solution under vigorous stirring. The reaction was allowed to proceed under vigorous stirring for 30 min. The formed polymer was filtered off and washed several times with distilled water and diethyl ether. The last traces of solvent were removed by keeping the polyamide in a freeze dryer for six hours.

### 3.4. Nanocapsules Characterization and Profiling

The produced Lys-PA and (FP-SAL/Lys-PA) NCs were further characterized using various techniques to develop an understanding of the properties of the produced materials.

#### 3.4.1. Fourier Transform Infrared (FTIR) Spectroscopy Analysis

The FTIR spectra of Lys-PA and (FP-SAL/Lys-PA) NCs were made using a Perkin Elmer FTIR spectrometer (PerkinElmer, Akron, OH, USA), coupled with Spectrum 10 software that was used to operate and treat the FTIR spectra. A few milligrams were placed on the lens and the FTIR scans were obtained over the range of 500–4000 cm^−1^ with a resolution of 2 cm^−1^.

#### 3.4.2. Nuclear Magnetic Resonance (NMR) Spectroscopy

A Bruker Avance DPX 300 spectrometer of 300-MHz/500-MHz (Bruker DPX-500) was employed to record the ^1^H-NMR (500 MHz) and ^13^C-NMR (125 MHz) of Lys-PA. Tetramethylsilane (TMS) was used as the internal standard (TMS representing 0.0 rpm). Chemical shifts were reported as parts per million (ppm).

#### 3.4.3. Differential Scanning Calorimetry (DSC) Analysis

The DSC analysis of Lys-PA and (FP-SAL/Lys-PA) NCs was carried out using a DSC Q200-TA instrument (TA Instruments, New Castle, DE, USA). Around a 2–3 mg sample was placed on an aluminum pan and heated at a rate of 10 °C per minute under the continuous purging of the nitrogen (50 mL/minute).

#### 3.4.4. Particle Size Analysis

A Zetasizer Nano ZS90 (Malvern Instrument, Malvern, UK) was used to measure the diameter of the average particle, the polydispersity index (PDI) and zeta potential of the Lys-PA and (FP-SAL/Lys-PA) NCs. Measurements were made at 25 °C where samples were diluted with deionized water before being analyzed and sonicated at medium amplitude (60 Hz) for 30 s. The analysis was performed in triplicate.

#### 3.4.5. High Performance Liquid Chromatography (HPLC) Assay Method for FP and SAL

Concurrent quantification of FP and SAL was conducted via a Dionex Softron HPLC system (Thermo Fisher Scientific Inc., Waltham, MA, USA), with a gradient pump, and a UV detector set to 239 nm at 25 °C. A Fortis C18 column (4.6 × 250 mm, 5 μm) (Fortis Technologies Ltd., Neston, UK) was used. The method employed was as follows: an isocratic method with a mobile phase comprising 70% acetonitrile and 30% 0.125% trifluoracetic acid (TFA) in water. The flow rate was 1 mL/min and the injection volume was 20 μL. Linearity was confirmed between 3.9 and 125 μg/mL. The International Conference on Harmonization guidelines for validation of analytical techniques was used to validate the method in terms of specificity, accuracy, precision, the limit of detection, and the limit of quantification [41].

#### 3.4.6. Drug Loading Capacity

The amount of loaded FP and SAL within the formulation was calculated according to two parameters: entrapment efficiency (*EE*), and encapsulated efficiency (*ECE*). *%EE* was calculated according to the following formula:(1)$$EE\;\left(\%\right)=\frac{API_{t\;}-API_{f}}{API_{t}}\times 100\;$$
where the “$API_{f}$” is the free FP or SAL in the supernatant, and “$API_{t}$” is the total amount of FP or SAL used to prepare the nanocapsules. The *ECE* was assessed based on measuring the content of FP and SAL in the final formula. About 10 mg of each formula was accurately measured and dissolved in 10 mL of methanol, filtered, and assayed using the validated HPLC method, and the % *ECE* was calculated according to Formula (2).
(2)$$ECE\;\left(\%\right)=\frac{API_{q}}{Formula_{t}}\times 100\;$$
where $TAPI_{q}$ is the weight of either FP or SAL in the tested formula, whereas ${\mathrm{Formula}}_{t}$ is the total weight of the produced formula (polymer and APIs).

#### 3.4.7. X-ray Diffraction (XRD) Analysis

X-ray diffraction of the APIs (FP and SAL) as well as the Lys-PA and (FP-SAL/Lys-PA) NCs were assessed using X-ray diffractometry (MiniFlex 600 benchtop diffractometer (RigaKu, Tokyo, Japan)). The XRD experiments were performed over the range 2θ from 5 to 99°, with Cu Kα radiation (1.5148227 Å) at a voltage of 40 kV and a current of 15 mA. All samples were fixed on a glass holder and scanned in triplicate. The data were recorded at a scanning speed of 5° per minute, and OriginPro^®^ software was employed to analyze the scans (OriginLab Corporation, Northampton, MA, USA).

#### 3.4.8. Scanning Electron Micrographs (SEM)

Surface characteristics of the (FP-SAL/Lys-PA) NCs were examined using JSM-IT300 (JEOL, Tokyo, Japan). About 2–5 mg of the polymer of the formula were sprinkled on a double-adhesive carbon tape affixed to an aluminium tub. Samples were analyzed without any further coating. ImageJ software was used to assess particle size. Particle size was measured manually using multipoint tools. The image was adjusted to show the color threshold, and clear aggregates with defined boundaries were selected and measured. Representative images are presented in Appendix A.

### 3.5. Assessment of the Performance of the (FP-SAL/Lys-PA) NCs

#### 3.5.1. Aerodynamic Performance Using a Next Generation Impactor (NGI)

The analysis of the aerodynamic performance of the final formula and in vitro deposition profile was established using the NGI (Copley Scientific Limited Model 170, Nottingham, UK). The NGI was connected to a high-capacity pump (model HCP5, Copley Scientific, Nottingham, UK) and a critical flow controller (model TPK 2000, Copley Scientific, UK). Size 3 gelatin capsules (Pharmacare, Amman, Jordan) were used to contain the formula. Approximately 20 mg of each formula was added per capsule, and six capsules were made each time to enable accurate quantification of the actives. Each capsule was placed into a DPI device (Aerolizer^®^) and dispersed into the NGI through the USP induction port at the flow rate of 60 L/min for 4 s per actuation to allow for 4 L of air at a pressure drop of 4 kPa. The NGI collection plates were conditioned with 1% *v*/*v* glycerin in acetone and allowed to dry for 30 min before use. Upon aerosolization, the deposited powder in each stage was dissolved with methanol, and the contents of the actives were assayed using the HPLC method as described earlier. The aerodynamic parameters, the mass median aerodynamic diameter (MMAD), the geometric standard deviation (GSD), the emitted dose (*ED)*, the fine particle fraction of the emitted dose (*FPF-E*), the fine particle fraction of the theoretical dose (*FPF-T*), and the respirable dose (*RD*), were calculated based on the dose deposited on the induction port (adapter and induction port), the pre-separator, stages 1 to 7, and the micro-orifice collector (MOC). The formulas used for the calculations of the aerodynamic parameters were as follows:(3)$$ED\;\left(\%\right)=\frac{\mathrm{Total}\;\mathrm{amount}\;\mathrm{of}\;\mathrm{API}\;\mathrm{from}\;\mathrm{induction}\;\mathrm{tube},\;\mathrm{preseprator}\;\mathrm{and}\;\mathrm{trays}\;1–7\;\&\;\mathrm{MOC}}{\mathrm{Total}\;\mathrm{amount}\;\mathrm{of}\;\mathrm{API}\;\mathrm{per}\;\mathrm{dose}}\times 100\;$$
(4)$$FPF-E\left(\%\right)=\frac{\mathrm{Total}\;\mathrm{amount}\;\mathrm{of}\;\mathrm{API}\;\mathrm{from}\;\mathrm{trays}\;2–7\;}{\mathrm{ED}}\times 100\;$$
(5)$$FPF-T\;\left(\%\right)=\frac{\mathrm{Total}\;\mathrm{amount}\;\mathrm{of}\;\mathrm{API}\;\mathrm{from}\;\mathrm{trays}\;2–7}{\mathrm{Theoretical}\;\mathrm{dose}}\times 100\;$$
(6)RD (μg)=total amount of API from trays 2–7 

As for the MMAD and GSD, they were calculated based on the USP technique <601> [46]. All results were carried out in triplicate and reported as mean ± standard deviation (SD).

#### 3.5.2. Release Study

The release profile of FP and SAL from the final formula was analyzed using a Microplate spectrophotometer Thermo Fisher, Finland. A total of 10 mg of the final formula (made in triplicate) for each active alone was placed in 2 mL of phosphate buffer saline (PBS) pH 6.8. The selected pH was to simulate the respiratory fluid alveoli pH of 6.9 ± 0.1, which was reported in the literature [47]. The wavelength was set at 239 nm for both FP and SAL. The device was set to record one reading every 15 min for 24 h. The cumulative amount of FP and SAL released into the solution was measured at pre-set time intervals at the corresponding λ-max (239 nm). The method was calibrated using PBS, and a calibration curve was constructed over a range of 3.9–62.5 µg/mL.

To develop a better understanding of the release pattern of FP and SAL for the nanoparticles, the release data were fitted into five kinetic mathematical models (zero-order release model, first-order release model, Higuchi model, Hixon Crowell model, and Korsmeyer–Peppas model). Equations (7)–(11) illustrate the models [42,48,49]:

Zero-order model:(7)$$API_{t}=API_{0}+K_{0}\times t\;$$

First-order model:(8)$$Log\;API_{t}=Log\;API_{0}-\frac{K_{1}\times t}{2.303}$$

Higuchi model:(9)$$API_{t}=K_{H}\times \sqrt{t}\;$$

Hixon Crowell model:(10)$$\sqrt[{3}]{API_{rt}}=k\times t\;$$

Korsmeyer–Peppas model:(11)$$\frac{API_{t}}{API_{\infty }}=K\times t^{n}\;$$
where $API_{t}$ is the amount of the active pharmaceuitcal ingredient (FP or SAL) that is released at time *t*, $API_{0}$ is the initial amount of active pharmaceutical ingredient (FP or SAL) (zero), $K_{0}$ is the zero-order rate constant, “*t*” is time, *K*_1_ is the first-order rate constant, *K_H_* is the Highuchi rate constant, $API_{rt}$ is the remaining amount of active pharmaceutical ingredient, “*k*” is Keppa (a constant), $API_{\infty }$ is the total amount of FP or SAL to be released, “*K*” is the release constant and “*n*” is the release exponent, (the value of “*n*” can be calculated by plotting the log percentage released versus log time—the slope of the line) [38,39,40].

### 3.6. Statistical Analysis

Data were analyzed with a one-way analysis of variance (ANOVA) and Tukey post-test using Minitab version 18. All the data were expressed as mean ± SD or relative standard deviation (RSD) and *p* < 0.05 was considered the level of significance.

## 4. Conclusions

In the present study, an FDC made of FP and salmeterol, which was loaded into a polymer made of a polyamide based on L-lysine nanocapsules, was successfully prepared using the interfacial polycondensation method. The morphological and molecular profiling of the nanocapsules supported their suitability for pulmonary drug delivery, as the nanocapsules produced aggregates in a favorable micron size of 1–5 µm. The in vitro assessment of the aerodynamic particle size distribution revealed that the produced nanoaggregates delivered almost 124 µg of FP from the prepared formulation in comparison with only 34 µg from the marketed product (Seretide Diskus^®^). Furthermore, it also delivered a nominated SAL FPF dose of 78% compared to 64% from the marketed product. The method used enabled the production of a simple, cost-effective process as an alternative option to the carrier-based formulation.

## Figures and Tables

**Figure 1 pharmaceuticals-15-00321-f001:**
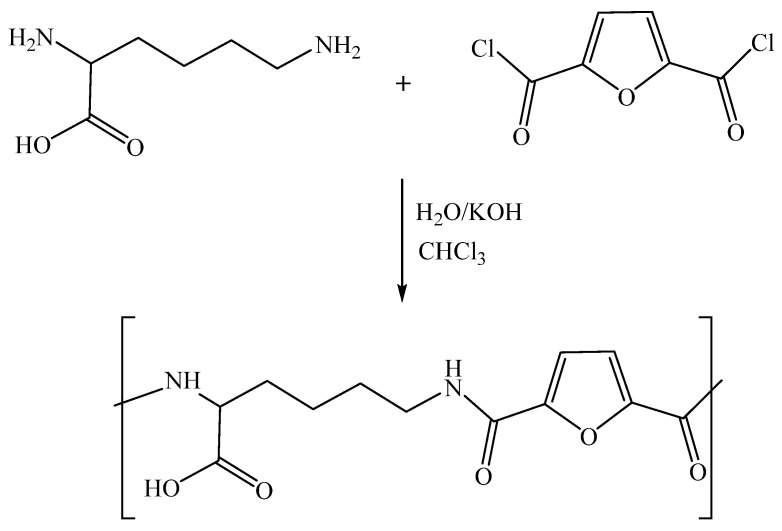
Synthesis of the polyamide by interfacial polycondensation of L-lysine and 2,5-furyl dichloride.

**Figure 2 pharmaceuticals-15-00321-f002:**
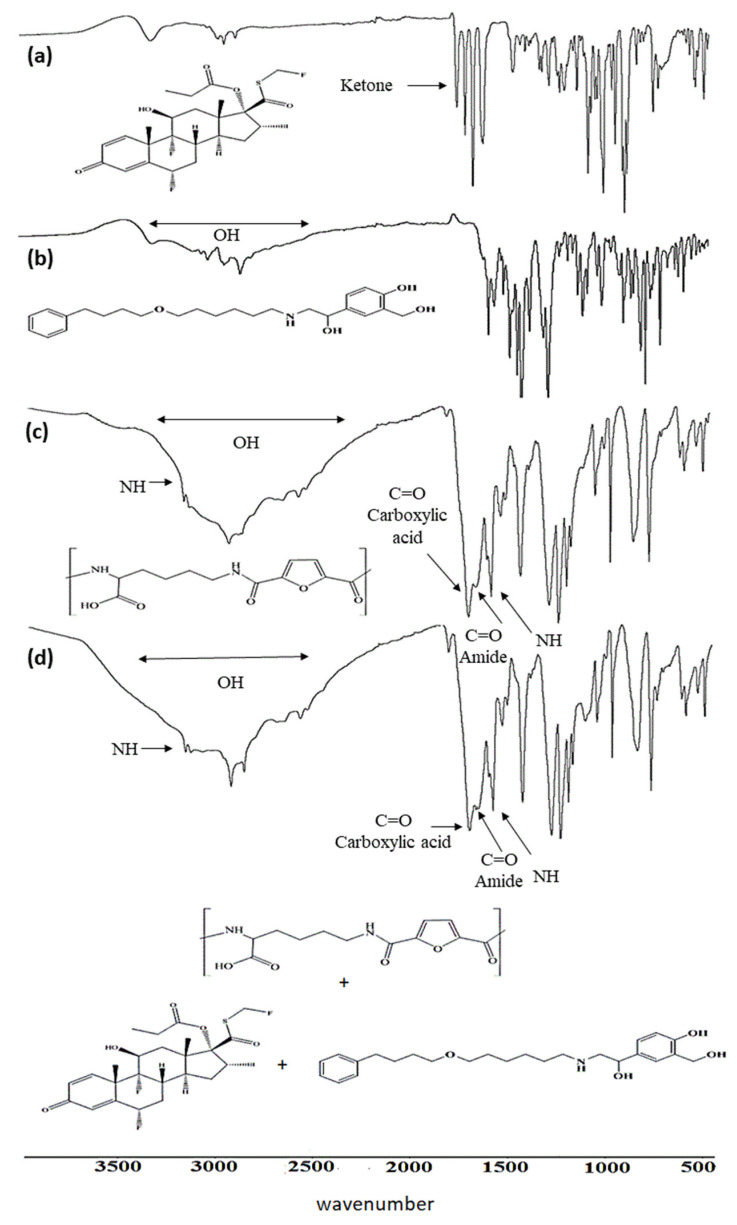
FTIR spectra of (**a**) fluticasone propionate (FP), (**b**) salmeterol xinafoate (SAL), (**c**) polyamide based on lysine, (**d**) and polyamide based on lysine formula.

**Figure 3 pharmaceuticals-15-00321-f003:**
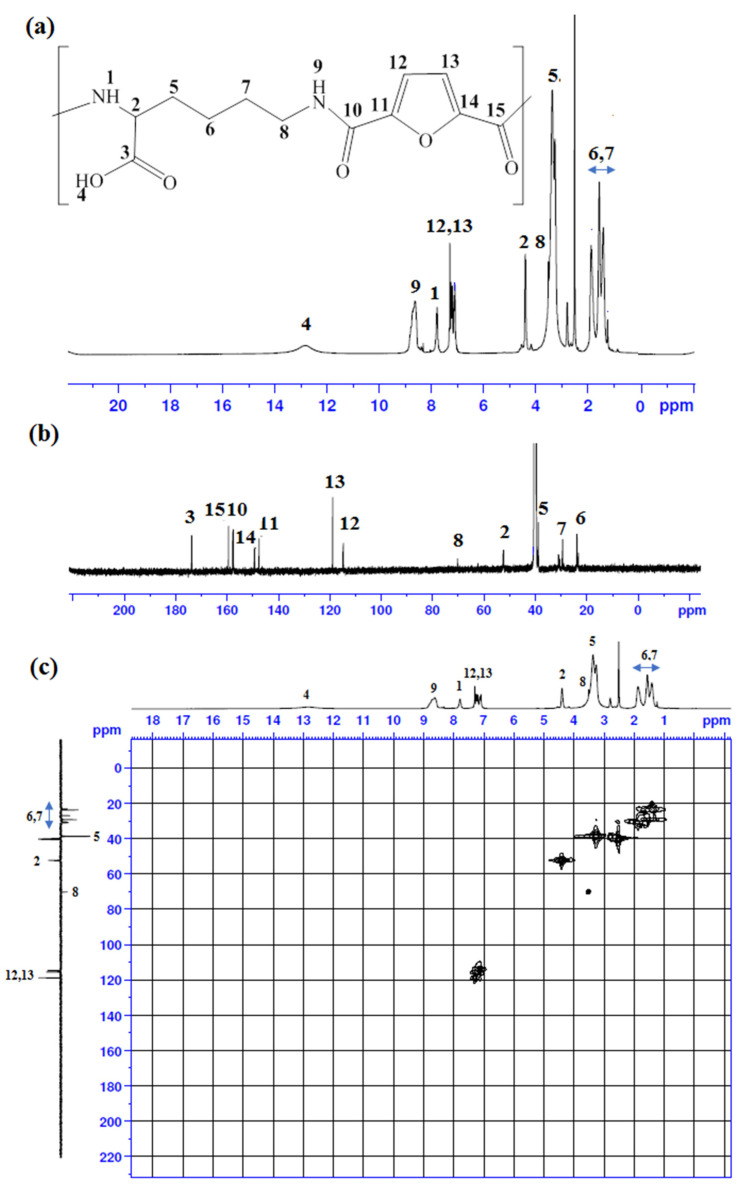
(**a**) ^1^H-NMR (**b**) ^13^C-NMR (**c**) DEPT 135-^1^H HMBC NMR spectra of (Ly-PA).

**Figure 4 pharmaceuticals-15-00321-f004:**
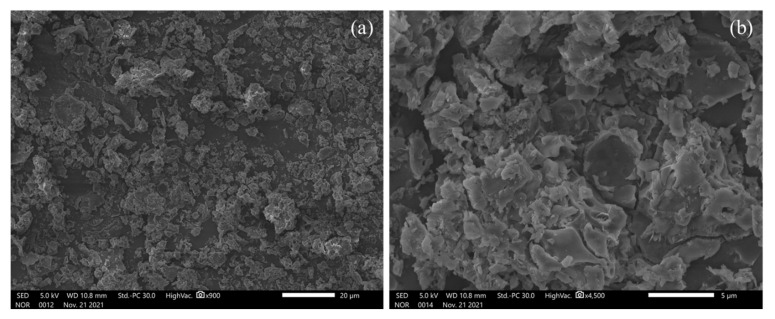
SEM micrographs of (FP-SAL/Lys-PA) NCs at (**a**) 900× magnification and (**b**) 4500× magnification, highlighting the aggregated nature of the NC.

**Figure 5 pharmaceuticals-15-00321-f005:**
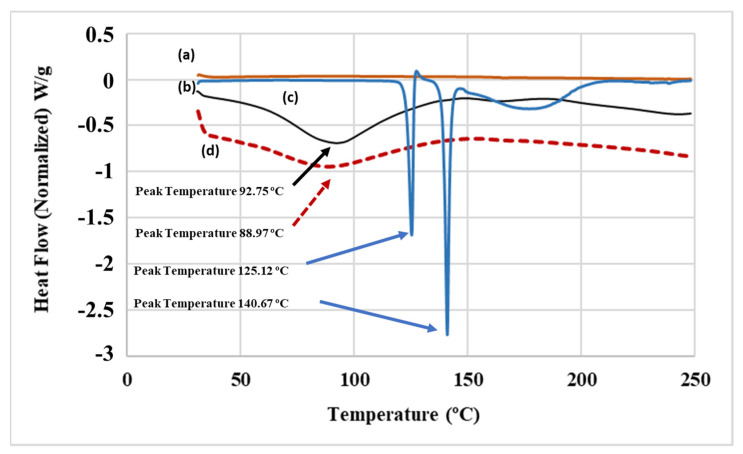
DSC curves for (**a**) fluticasone propionate (FP), (**b**) salmeterol xinafoate (SAL), (**c**) (FP-SAL/Lys-PA) NCs and (**d**) Lys-PA.

**Figure 6 pharmaceuticals-15-00321-f006:**
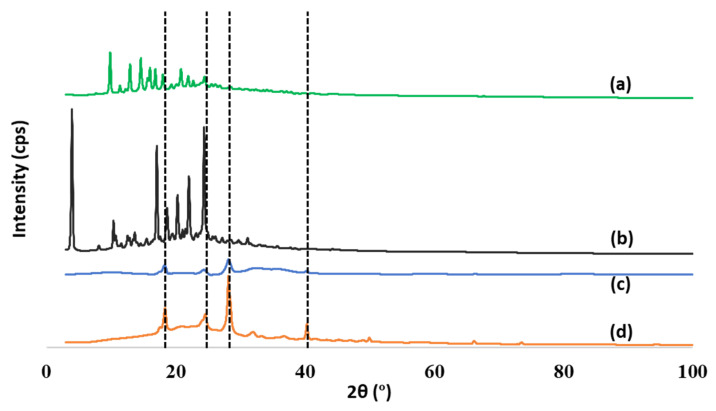
Powder XRD patterns: (**a**) FP, (**b**) SAL, (**c**) (FP-SAL/Lys-PA) NCs and (**d**) Lys-PA.

**Figure 7 pharmaceuticals-15-00321-f007:**
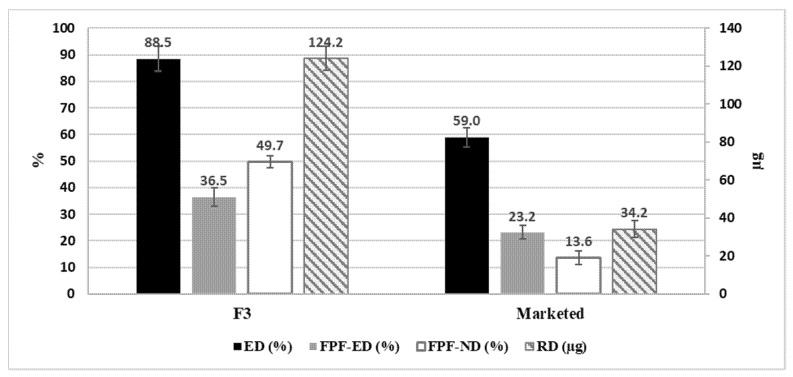
Aerodynamic behavior of FP from Formula F3 containing (FP-SAL/Lys-PA) NCs, highlighting the main parameters for the formulation. ED: emitted dose; FPF-ED: fine particle fraction of emitted dose; FPF-ND: fine particle fraction of nominated dose; RD: respirable dose. Results are presented as mean ± SD, *n* = 3.

**Figure 8 pharmaceuticals-15-00321-f008:**
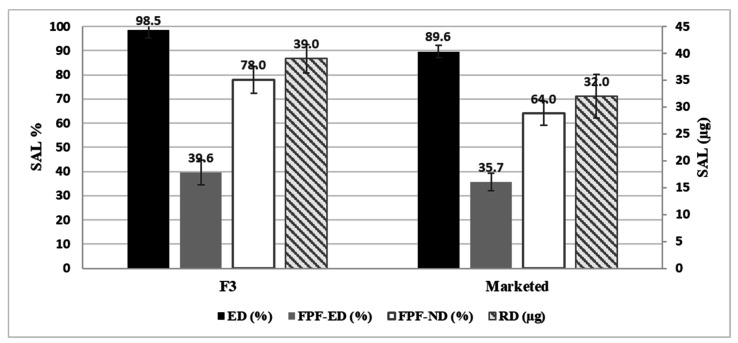
Aerodynamic behavior of SAL from Formula F3 containing nanocapsules, highlighting the main parameters for the formulation. ED: emitted dose; FPF-ED: fine particle fraction of emitted dose; FPF-ND: fine particle fraction of nominated dose; RD: respirable dose. Results are presented as mean ± SD, *n* = 3.

**Figure 9 pharmaceuticals-15-00321-f009:**
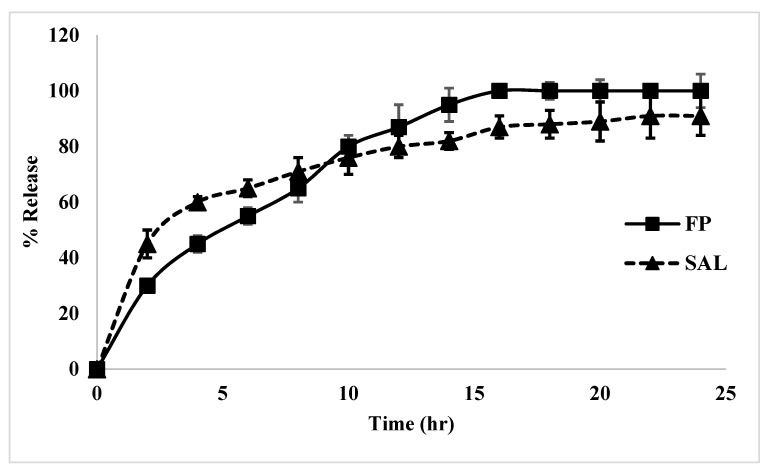
**The** release profile of FP and SAL from FP-SAL-loaded polymer nanocapsules (F3) for 24 h (mean ± SD, *n* = 3).

**Table 1 pharmaceuticals-15-00321-t001:** Particle size analysis of fluticasone propionate (FP), salmeterol xinafoate (SAL), the polymer and FP-SAL-polymer loaded nanocapsules (NC) (mean ± SD, *n* = 3).

	Material	FP	SAL	Polymer	FP-SAL-Polymer Loaded NC
Parameter					
Particle size (nm)		2230.7 ± 200.3	1935.7 ± 188.9	199.3 ± 12.5	226.7 ± 35.3
Polydispersity index		0.91 ± 0.15	0.57 ± 0.15	0.38 ± 0.21	0.32 ± 0.08
Zeta potential (mV)		−27.2 ± 5.8	−18.5 ± 0.5	−28.16 ± 1.6	−30.6 ± 4.2

**Table 2 pharmaceuticals-15-00321-t002:** Validation parameters of FP and SAL using the HPLC method.

Parameter	FP	SAL
Calibration curve equation	y1=0.7011x+0.6237 $\left(R^{2}:0.9999\right)$	y2=0.5092x−0.941 $\left(R^{2}:0.9999\right)$
Limit of detection (LOD) µg/mL	2.04	0.745
Limit of quantification (LOQ) µg/mL	6.18	2.26
Intraday % Recovery (mean ± SD) (*n* = 3)		
• 125 µg/mL	102.85 ± 4.69	100.04 ± 3.74
• 31.25 µg/mL	99.58 ± 4.25	99.69 ± 4.32
• 3.9 µg/mL	97.94 ± 3.02	100.15 ± 4.96
Interday % Recovery (mean ± SD) (*n* = 3)		
• 125 µg/mL	98.54 ± 5.69	102.57 ± 4.27
• 31.25 µg/mL	101.58 ± 4.98	97.27 ± 5.07
• 3.9 µg/mL	98.98 ± 6.98	103.07 ± 4.27
Precision (concentration 62.5 µg/mL) mean ± SD, RSD) (*n* = 10)	98.76 ± 2.05, 2.08%	101.24 ± 3.22, 3.18%

**Table 3 pharmaceuticals-15-00321-t003:** API of (FP-SAL/Lys-PA) NCs with composition, *EE*%, *ECE*%, recommended formula weight, *EE*% and FPP%.

	Formula	F1 (Single API)	F2 (Single API)	F3 (Double APIs)
Parameter				
FP (mg, mmol)		10	20	20
SAL (mg)		5	5	2
*EE* (%) FP-SAL		91.6–91.2	81.13–92.5	79.8–99.17
*ECE* (%) (FP-SAL)		1.03–0.63	1.66–0.702	1.67–0.33
Recommended formula weight (mg) (FP-SAL)		24.3–7.9	15.1–7.1	15

**Table 4 pharmaceuticals-15-00321-t004:** MMAD and GSD of the (FP-SAL/Lys-PA) NCs and the marketed product.

	Formula	F3		Marketed	
Parameter		FP	SAL	FP	SAL
MMAD (µm)		1.559	1.778	1.434	1.524
GSD		0.080	0.047	0.074	0.081

**Table 5 pharmaceuticals-15-00321-t005:** The coefficients of determination (R^2^) as well as the “*n*” value for each mathematical model for FP and SAL.

Mathematical Model	FP-R^2^	SAL-R^2^	FP-n	SAL-n
Zero order	0.9824	0.9326		
First order	0.9231	0.8744		
Higuchi model	0.9935	0.9722		
Hixon Crowell model	0.8872	0.9355		
Korsmeyer–Peppas model	0.999	0.9838	0.5541	0.3176

## Data Availability

Data is contained within the article and Appendix A.

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
