# Peer review of "Novel Fluticasone Propionate and Salmeterol Fixed-Dose Combination Nano-Encapsulated Particles Using Polyamide Based on L-Lysine"

_pharmaceuticals, 2022, doi:10.3390/ph15030321_

Round 1
Reviewer 1 Report
The manuscript “Novel Fluticasone Propionate and Salmeterol Fixed-Dose Combination nano-encapsulated particles using polyamide based on L-lysine” is designed and written well. The manuscript needs some edits before acceptable for publication.
The manuscript must be verified for typo and syntactic errors.
Abstract
Define DPIs in the abstract
Hence, an appropriate approach that retains the properties of the actives and produces the targeted performance is required. This statement not necessary in the abstract.
Write the units of ZP.
How the DSC, XRD and SEM confirms the molecular profiling?
Write the marketed product used as control.
Main text
The abbreviations not defined in the manuscript eg. FDC, LABAs.
Further, FDCs have the potential to improve and prolong the effect of monotherapy. Why the combination needed if monotherapy showed better therapeutic activity.
What is need of the FDC of selected drugs. Clarify in the manuscript.
Why the L-lysine based amino acid used for the nanoaggregates preparation. Explain.
The interfacial polycondensation procedure also prepared FP and salmeterol xina-foate. This statement very unclear.
This direct method of encapsulation of FP and SAL resulted in capsules with an average diameter of around 199.3 nm and high-efficiency drug encapsulation of 19.8%. how did the EE of both drugs reported as 19.8%. write the EE of each drug in the nanocapsule separately.
Write the reference for FTIR of lysine.
Figure 2 – use full names of the FP and SAL.
Delete Pdi from table 1.
Author Response
Dear Reviewer
Many thanks for your valuable comments. Attached please find the response for each comment. Also, we added a supplementary file with all required data and an updated manuscript.
Best regards,

Reviewer 2 Report
The article titled “Novel Fluticasone Propionate and Salmeterol Fixed-Dose Combination nano-encapsulated particles using polyamide based on L-lysine” may be a useful contribution to the journal; however, I recommend a major revision should be taken into consideration before publishing the work.
- Please provide the abbreviations or use them appropriately. Eg., on Page number 1, line number 22 the author mentioned DPIs, but the explanation was not given in the manuscript. If it is a dry powder inhaler, then please change to DPI instead of DPIs.
- The precise information about the procedure in the methodology section is highly recommended. In Section 3.3, please include the required quantity of FP and SAL, used for FP-SAL/Lys-PA NCs synthesis.
- In table 1, please use PDI instead of Pdi.
- The author mentioned that the zeta potential of synthesized FP-SAL- polymer loaded NC is -30.6 ±2 and the SEM images (Figure 4) show the aggregated nature of the NC. According to the zeta potential results (above -25), the synthesized final materials are strongly stable system and free from aggregation. Please explain the reason for aggregation in the SEM results. SEM images must be taken for unloaded NC (Lys-PA) as well.
- Please explain how the average aggregates are calculated as 3.51 ± 1.21 µm. Please provide the ImageJ processed SEM image in the supplementary section to support your claim.
- Please provide the graphical results of particle size and zeta potential in the supplementary information.
- The SEM and Zetasizer (Particle size) results are highly contradictory, I strongly recommend analysing the sample using Transmission Electron Microscopy to bring out a better conclusion on the size of the synthesized materials.
- In Figure 5, the author mentioned that sample (a) belongs to Lys-PA, and it is believed that it is an unloaded material (without FP and SAL), then how does the author denotes (c) and (d) in the same curve. The representation in figure 5 in section 2.6 is totally contradictory. Please make the necessary changes and modify the discussion accordingly.
- There was no proper discussion given in section 2.6. How the author concluded a small endothermic peak appeared in 169.96 is belonged to SAL. I strongly recommend the author perform DSC experiments for SAL and FP separately and add them into Figure 5 and modify the discussion accordingly.
- Please include the formula to calculate ECE in Section 3.4.6.
- Please rewrite section 3.5.2, the present version is not so clear enough. The author mentioned that the wavelength was set at 239 nm, and mentioned that the cumulative amount of FP and SAL released into the solution was measured at pre-set time intervals at the corresponding λ-max, HOW? Please give the exact wavelength used to measure the release of both fluticasone propionate and salmeterol xinafoate. Please state why the pH was maintained at 6.8 instead of the physiological pH of 7.4.
- In Section 2, the author only gives the results and explained them. There was no proper discussion was given by adding any suitable references to support their claims in each characterization study. I strongly recommend modifying the discussion in Section 2.
Author Response

(The authors gave the same response as above.)

Reviewer 3 Report
This research introduced a one-step approach to expedite the synthesis of nanoaggregates made from a biocompatible and biodegradable polyamide based on L-lysine amino acid. Two APIs including fluticasone propionate and salmeterol xinafoate were explored in this research. The fine particle fraction of the nominated dose was superior to the marketed product. The results suggest that nanoaggregates using polyamides based on L-lysine and interfacial polycondensation can serve as a good platform for pulmonary drug delivery of FDC systems. While there are several critical points for the analysis that should be clarified. Before considerating the publication, please respond to the following questions:
- In the FTIR analysis part, the assignment of the functional groups should be talked about or at least give the reference to analyze the absorption bands.
- In the section of particle size analysis, how many groups of experiments were studied to give the range of the size, like ± 200.3, ± 188.9, etc.? The usage of imageJ should be detailed, for example, how to distinguish the single-particle, overlap particle, and agglomerate particle?
- In the DSC analysis section, the discussion is not matched Figure 5-a, 5-b (caption), etc., please check the discussion section. A wide-range DSC test is recommended. By the way, the unit of the x-axis should be degrees Celsius.
- Page 12, line 281, the author analyzed part reason for the release is from the biodegradation of the polymer. The degradation time during the release time or the kinetics of biodegradation should be clarified to assist to analyze the release performance.

Author Response

(The authors gave the same response as above.)

Round 2
Reviewer 1 Report
The manuscript improved as per the suggested edits for more clarity. No further comments from my end.
Reviewer 2 Report
The authors have replied to almost all the comments, and it has been improved in the revision. I recommend considering for publication in “MDPI-Pharmaceuticals”